# Trigger Points of Necroptosis (RIPK1, RIPK3, and MLKL)—Promising Horizon or Blind Alley in Therapy of Colorectal Cancer?

**DOI:** 10.3390/ijms262211101

**Published:** 2025-11-17

**Authors:** Marcin Sokołowski, Aleksandra Butrym

**Affiliations:** 1Dr Alfred Sokołowski Specialist Hospital in Walbrzych, 58-309 Walbrzych, Poland; aleksandra.butrym@gmail.com; 2Department of Hematology and Oncology, Wroclaw Medical University, Branch in Walbrzych, 50-556 Wroclaw, Poland; 3Oncology Clinical Trial Support Center, Alfred Sokolowski Specialistic Hospital in Walbrzych, 58-309 Walbrzych, Poland

**Keywords:** necroptosis, colorectal cancer, RIPK1, RIPK3, MLKL

## Abstract

Colorectal cancer (CRC) remains one of the leading causes of cancer-related mortality worldwide, due to the limited efficacy of current therapeutic strategies in advanced stages. Necroptosis, a regulated form of necrotic cell death mediated by receptor-interacting protein kinases RIPK1 and RIPK3, and the pseudokinase MLKL, has emerged as a potential alternative pathway to induce cancer cell death. Recent studies suggest that modulation of necroptosis may enhance antitumor immunity, overcome therapeutic resistance, and improve clinical outcomes in CRC. In this review, we systematically analyzed the current literature on the role of necroptosis in CRC, focusing on molecular mechanisms, experimental models, and therapeutic implications. By critically evaluating the available evidence, we aimed to determine whether targeting RIPK1, RIPK3, and MLKL, or other novel agents, represents a promising horizon or a blind alley in the development of novel CRC therapies.

## 1. Introduction

Colorectal cancer is the third most common cancer and the second leading cause of cancer-related deaths around the world. The National Center for Health Statistics has estimated 153,020 cases and 52,550 deaths from colorectal cancer in the USA. Every year approximately 1.8 million new cases are diagnosed and about 880,000 people die because of colorectal cancer [1,2]. The epidemiology of CRC depends on age, gender, world region, and racial group. The overall annual age-standardized CRC incidence rate in the USA has decreased by 46% in 2019 since its peak in 1985 [1]. This decline is associated with changing in lifestyle, for example, restrictions on smoking and the common use of nonsteroidal anti-inflammatory drugs [1]. Furthermore, mass colonoscopy screening led to an increased proportion of CRC diagnosed in the localized stage disease. The percentage increased from 33% in 1995 to 41% in 2005 [1]. The introduction of endoscopic screening resulted in increased overall 5-year relative survival from 50% in the mid-1970s to 65% during the period 2012–2018. Generally, the stage at diagnosis is the most important predictor of survival, with 5-year relative survival ranging from 91% for localized disease to only 14% for distant disease [1]. The novel agents based on new molecular pathways, could definitely change these unfavorable statistics and extend the overall survival in the population of patients with advanced CRC. New branches of therapy are focused on the following: monoclonal antibodies targeted EGFR (epidermal growth factor receptor), such as cetuximab [3] and panitumumab [4], or VEGF (vascular endothelial growth factor) agents, including aflibercept [5], bevacizumab [6,7], and regarofenib [8]; new chemotherapy agents, trifluridine and tipiracil hydrochloride [9]; and immune checkpoint inhibitors like pembrolizumab [10] or nivolumab with ipilimumab [11,12].

Despite significant progress of advanced colorectal cancer therapy, the effects of treatment are still insufficient. Contemporary oncology is still looking for new trigger points and prospective branches of therapy. One of the most promising ways involves the programmed cell death pathway, necroptosis. Despite its etymology, it is not a simple combination of necrosis and apoptosis. This unique cell death mechanism is characterized by a series of morphological and biochemical features, including cell membrane rupture, organelle swelling, cytoplasmic and nuclear decomposition, and the release of DAMPs [13]. The role of necroptosis has been proven in liver diseases [14], kidney injuries [15], neurodegenerative disorders [16], cardiovascular diseases [17], and multiple types of malignancies such as breast cancer [18], multiple myeloma [19] and non-small cell lung cancer [20]. Below, we would like to present a brief history, a mechanism description, and the potential role of necroptosis in colorectal cancer.

## 2. Natural History of Necroptosis

The initial discovery of a new programmed death cell pathway was an observation performed in 1988, which indicated that tumor necrosis factor (TNF) could induce not only apoptosis but also necrosis in certain cell lines [21]. In 1998, Vercammen et al. proved that inhibition of caspases increases the sensitivity of L929 cells to necrosis mediated by TNF [22]. In the same year, Kawahara et al. proposed a new type of Fas-mediated necrosis, induced by Fas-associated protein with death domain (FADD) with a blockade of the apoptotic mechanism [23]. Regardless of the fact that researchers could not identify a new pathway, they continued to look for a new way of cell death. Although researchers had long been unable to identify a novel cell death pathway, continued investigation led to the discovery in 2000, by Holler et al., of a distinct mechanism of programmed cell death induced by tumor necrosis factor-α (TNF-α), tumor necrosis factor-related apoptosis-inducing ligand (TRAIL), or Fas ligand (FasL) [24]. In 2003, Chan and colleagues further described a new mechanism of “programmed necrosis” based on RIPK1 [25].

Subsequently, the theory of “programmed necrosis “evolved with the identification of necrostatin-1 (NEC-1) as an inhibitor that acts through interaction with RIPK1 [26]. In 2009, HE et al. described receptor-interacting protein kinase 3 (RIPK3) as a downstream regulator of RIPK1-dependent programmed necrosis [27]. Afterwards, Sun L. and colleagues described the mixed lineage kinase domain-like pseudokinase (MLKL) as a downstream substrate of RIPK-3 in TNF-induced “programmed necrosis” [28]. In 2012, the Nomenclature Committee on Cell Death (NCCD) recommended “necroptosis” as a term to describe the novel type of cell death [29]. Despite this recommendation, the final definition was provided by NCCD in 2018, describing necroptosis as a modality of regulated cell death (RCD) triggered by perturbations of extracellular or intracellular homeostasis that critically depends on MLKL, RIPK3, and RIPK1 [30]. It is also worth noting the discovery reported in 2013 that toll-like receptors (TLRs) can induce necroptosis through a TRIF–RIPK3–MLKL signaling axis, functioning as a pathway independent of death receptor activation [31]. The history of discoveries connected with necroptosis is presented in Figure 1.

## 3. Molecular Mechanism of Necroptosis

The most extensively characterized mechanism of necroptosis is a TNF-α- dependent pathway [32]. Tumor necrosis factor-α (TNF-α) binds to its receptor, tumor necrosis factor receptor 1 (TNFR1), which is typically expressed as a trimer on the cell surface. Upon ligand binding, TNFR1 undergoes conformational changes that enable the recruitment of adaptor proteins to its cytosolic death domain. These include TRADD, RIPK1, TRAF2, TRAF5, and cIAP1/2, forming the membrane-bound complex I. The ubiquitination of RIPK1 by cIAP1/2 stabilizes complex I and facilitates the recruitment of the IκB kinase (IKK) complex (IKKα, IKKβ, and NEMO), leading to NF-κB activation and pro-survival signaling [33,34,35]. The deubiquitination of RIPK1 promotes complex II formation, comprising TRADD, FADD, caspase-8, and RIPK1. Interaction between RIPK1 and RIPK3 within this complex can trigger necroptosis, depending on caspase-8 activity and the cellular context [36,37].

A key regulatory checkpoint in programmed cell death is the balance between apoptosis and necroptosis, largely governed by caspase-8. Active caspase-8 cleaves and inactivates RIPK1 and RIPK3, suppressing necroptosis and promoting apoptosis. Inhibition or loss of caspase-8 activity—either genetically (e.g., CASP8 mutations) or pharmacologically (e.g., Z-VAD-fmk)—shifts this balance toward necroptosis [38,39,40].

The second condition is the deubiquitination of RIPK1, influenced by multiple factors in cytosol, such as cIAP1, cIAP2, and X-linked inhibitor of apoptosis (XIAP) [41]. On the other hand, the deubiquitination agents, RIPK1-deubiquitinating enzyme cylindromatosis (CYLD), protein A20 (TNFAIP3-ubiquitin-modifying TNF alpha-induced protein 3), or linear ubiquitin chain assembly complex (LUBAC), are also present in the cytosol [42,43,44]. The activation of these deubiquitination factors leads to phosphorylation of RIPK3 by RIPK1. RIPK1 and RIPK3 interact through RIP homotypic interaction motif (RHIM), which leads to trans- and autophosphorylation, involving heat shock protein 90 (HSP90) and cell division cycle 37 homolog (CDC37) [45]. As a result, they form a micro-filament-like complex called necrosome and activate mixed linage kinase domain like pseudokinase (MLKL) [46]. Full activation of MLKL is a result of the phosphorylation of amino acids in all possible positions (S124, S158, S228, and S248) [47]. The dephosphorylation of RIPK3 and the phosphorylation of MLKL in position T357 and S358 are catalyzed by protein phosphatase Mg^2+^/Mn^2+^-dependent 1B (PPM1B) and protein Chip (Stub1) [48,49]. The phosphorylated protein presents the intracellular part (N-terminus of four helical proteins–NB Domain) and brace region, which leads to the oligomerization of MLKL, with the participation of HSP90. The modified conformation allows phosphatidylinositol phosphate (PIP) to bind on the inside of the cell membrane [50]. Even though the initial phase of binding takes place in the brace region and leads to its translocation to the cell membrane, the NB domain is also changed by the rolling-over process, to expose PIP-binding sites. Sequentially, the central brace region changes its position and stabilizes the binding [51]. The next stage of necroptosis is not clear. MLKL leads to permeabilization of the cell membrane, which could be connected with uncontrolled Ca^2+^ influx by TRPM7 receptor [52]. Other trials indicated a fragmentation of the mitochondrial membrane by PGAM5 and DNM1L [53]; however, this theory has been recently rejected [54]. The consequence is a release of the alarmins-damage-associated nuclear pattern (DAMP) proteins, which are characteristic of necrosis. The production of alarmins, e.g., extracellular nucleic acids, ATP, protein HMGB2, crystals of urine acid, and heat shock proteins, makes a necroptosis similar to necrosis [55].

Other mechanisms of necroptosis initiation should also be mentioned—for example, TNF-related apoptosis-inducing ligand (TRAIL) which activates DR4/5 and binds FADD to form complex IIb, which recruits and activates caspase 8, resulting in apoptosis. In the case of the absence of caspase 8, the necroptotic pathway is promoted [56].

Necroptosis can also be initiated via toll-like receptors (TLRs) located on the cell membrane surface, which are activated by extracellular dsRNA and lipopolysaccharide (LPS). Upon activation, TLRs recruit the TIR-domain-containing adaptor-inducing interferon-β (TRIF), which interacts with RIPK1. The autophosphorylation with RIPK3 leads to MLKL activation and initiates necroptosis. The RIPK3 could also be activated directly by TLR4 and TLR3, independently of RIPK1 [57]. The molecular pathway of necroptosis is presented in Figure 2.

Another pathway is based on viral z-RNA/DNA produced during replication, which activate z-DNA binding protein 1 (ZBP1) in the nucleus. ZBP1 leads to the induction of RIPK3 and MLKL, and, subsequently, necroptosis [58,59].

Interferon is also capable of inducing necroptosis by activating the RIPK1-RIPK3 complex via the JAK/STAT signaling pathway in the case of unphosphorylated/absent FADD [60]. Additionally, interferon can recruit RIPK3 and MLKL to necroptosis pathways by ZBP [61].

Searching for the new branch of therapy could be achieved by controlling the trigger point of necroptosis. In our review, we focused on three central effectors: MLKL, RIPK1, and RIP3. Inhibition or stimulation of these proteins may affect the entire necroptotic process and potentially alter natural course of colorectal cancer.

## 4. RIPK 1—Receptor-Interacting Serine/Threonine Kinase 1

RIPK1 is a crucial enzyme involved in the necroptosis and apoptosis process [62]. In the human genome, it is encoded by the RIPK1 gene, which is located on chromosome 6 on loci 6p25.2. It belongs to the Receptor-Interacting Protein (RIP) kinases family and is composed of 671 amino acids with molecular weight of approximately 76 kDa. The protein contains three domains: a kinase domain, a death domain, and an intermediate domain [63,64]. The first one is the serine/threonine kinase domain located on the 300 aa N-terminus, which interacts with Necrostatin-1 [65] or TRAF-2 [66]. The death domain is situated on the 112 aa C-terminus and may ligate Fas, TRAILR2 (DR5), and TNFR1 [67]. The intermediate domain is crucial for NF-kB activation and RHIM-dependent signaling. It can also interact with TRAF2, NEMO, RIPK3, ZBP1, and OPTN [68].

The multipotential character of RIPK 1 can be divided into cell death processes (necroptotic and apoptotic) and its pro-survival role. The kinase is a component of protein complex I, which also includes TRADD, TRAF2, TRAF5 (TNF receptor-associated factor 2 and 5), cIAP1, and cIAP2 [31,32]. Complex I can be modified by the Inhibitor of Apoptosis Proteins (IAP) and the Linear Ubiquitination Assembly Complex (LUBAC) through ubiquitination, which leads to activation of the NF-ƒΘB. Subsequently, it activates the expression of FLICE-like inhibitory protein (FLIP), which binds to caspase-8 and forms a heterodimer in the cytosol. Dimerization inhibits caspase-8 mediated apoptosis and leads to form complex IIb (caspase-8 FLIP heterodimer with RIPK1 and RIPK3). Caspase inhibition within this complex allows RIPK1 and RIPK3 to autotransphosphorylate each other, transforming the complex into a necrosome. The necrosome recruits Mixed Lineage Kinase Domain-Like protein (MLKL), which is phosphorylated by RIPK3 and rapidly translocates to lipid rafts inside the plasma membrane. Emergent pores in the membrane allow sodium influx, which changes the osmotic pressure, leading to cell membrane rupture.

On the other hand, RIPK1-mediated pathways can also influence the expression of NF-kB, a protein complex known to regulate transcription of DNA, which regulates the survival processes [69].

The upregulation and overexpression of RIPK1 have been confirmed in human colorectal cancer cells. RIPK1 interacts with mitochondrial Ca^2+^ uniporter (MCU), which promotes proliferation by increasing the mitochondrial Ca^2+^ uptake and energy metabolism. The ubiquitination site of RIPK1 (RIPK1-K377) appears to be critical for this interaction with MCU and for the function promoting cell proliferation. Furthermore, RIPK1-mediated cell proliferation through MCU is a central mechanism underlying colorectal cancer progression. This path could be a therapeutic target for novel agents [70]. The fragile X messenger ribonucleoprotein (FMRP) binds RIPK1 mRNA and regulates necroptosis process [71].

The RIPK1 also interacts with TRAF-6 through the polyubiquitination of Lys48-linked RIPK1, which leads to a reduced level of kinase in colorectal cancer cells. The breaking death pathways balance towards necroptosis, leading to the proliferation of neoplasmatic cells [72]. Kang et al. postulated a control role performed by RIPK1 to enhance metastasis of CRC via WNT/β-catenin canonical signaling. The team has been working on human colorectal cancer cell lines, HCT116 and DLD-1, and have proven the overexpression of RIP1 and β-catenin after the WNT3A treatment. Subsequently, they injected intravenously the overexpressed CRC cell, which demonstrates the role of RIP1 in the promotion of metastasis. The investigation suggests that WNT3A treatment induces direct binding between RIP1 and β-catenin. The process leads to stabilization of the β-catenin which is connected with RIP1 ubiquitination via the downregulation of E3 ligase and cIAP1/2. Furthermore, the elimination of cIAP1/2 expression in vitro stimulates the expression of RPK-1 and β-catenin. As a result, the CRC cells tend to migration and invasion via the stimulation of EMT (endothelial–mesenchymal transition) [73]. We summarized the findings of role of RIPK1 in CRC in Table 1.

## 5. RIPK 3—Receptor-Interacting Serine/Threonine Kinase 3

Receptor-interacting serine/threonine kinase 3 is the next crucial protein for the necroptosis process. As an RIP family member, it contains a unique C-terminal. The encoded protein is predominantly localized in the cytoplasm, and it also can export signals to the nucleocytoplasmic pathway. RIPK-3 is a part of the tumor necrosis factor (TNF) receptor-I signaling complex, and can induce apoptosis and necroptosis [74].

Expression of RIPK3 is probably connected with prognosis in metastatic colorectal cancer. A small study suggested that progression-free survival was significantly shorter in low-expression group of RIPK3 than the cohort with its high expression (5.6 months vs. 8.4; *p* = 0.02). Moreover, patients with the high expression of RIPK3 were associated with a lower risk of disease progression (HR 0.61, 95% CI, 0.38–0.97; *p* = 0.044) and had significantly longer overall survival (OS) (29.3 months vs. 18.5 months, *p* = 0.036). The univariate analysis of that study showed that a high level of RIPK3 expression was associated with a longer OS (HR 0.59; *p* = 0.044) [75].

Overexpression of RIPK 3 was also observed in mouse cell models of colorectal cancer (CRC) and in a subset of human CRC cell lines [76]. In contrast to the mentioned observation, tumors in patients with inflammatory bowel diseases had a lower expression of RIPK3, which was connected with progression of the disease [77]. Another in vitro study was focused on role of RIPK3 in colitis-associated tumorogenesis. The samples were obtained from tissues, from colorectal cancer patients and mice cells, where the colitis-associated cancer was induced using an azoxymethane injection followed by dextran sodium sulfate treatment in RIP3-deficient or wild-type mice cells. The analysis showed that RIPK3 is upregulated in mouse colitis-associated cancer cells and in human colon cancer cells. The deficiency of RIPK3 in mice cells was connected with the inhibition of colitis-associated tumorogenesis. The crucial role in tumorogenesis is by the immunosuppressive microenvironment, which plays a role in the expression of chemokine CXCL1, which is induced by RIPK3. In conclusion, authors suggest that RIP3 could enhance the proliferation of premalignant intestinal epithelial cells (IECs) via JNK (c-Jun N-terminal kinase) signaling [78]. We summarized findings of role of RIPK3 in CRC in Table 2.

## 6. MLKL—Mixed Lineage Kinase Domain-Like Pseudokinase

MLKL is a member of kinase superfamily, characterized by protein kinase-like domain. The protein is encoded on the MLKL gene, located on chromosome 16 at locus 16q23.1, composed of 471 amino acids with molecular mass of 54479 Da. As we mentioned, it plays a crucial role in TNF-induced necroptosis via interaction with RIP3 [79]. The first characteristic of the crystal structure of MLKL was proposed by Murphy et al. in 2013. They postulated that MLKL is composed of an N-terminal four-helix bundle (4HB, residues 1–117), a two-helix linker (brace domain, residues 129–169), and a C-terminal pseudokinase domain (residues 171–464). They also categorized MLKL as a “pseudokinase” due to the absence of two of three crucial catalytic residues that are typically conserved in protein kinases and are essential for phosphoryl transfer activity [80].

The role of mixed lineage kinase domain-like protein in the carcinogenesis of CRC has been described in several studies. MLKL takes part in necroptosis as a critical mediator, which results in the release of cellular damage-associated molecular patterns (DAMPs). In mice cell lines, the knockout of Mlkl led to enhanced colitis and colitis-associated tumorigenesis (CAT), which was associated with massive leukocyte infiltration and increased inflammatory responses. Intestinal mucosal tissue and polyps isolated from Mlkl-/- mice exhibited increased ERK activation and an elevated expression of genes associated with inflammation and cancer. Mechanistically, the enhanced inflammation in Mlkl-/- mice was attributed to MEK/ERK activation, particularly in dendritic cells (DCs) [80]. Similarly to other crucial necroptotic proteins, the expression of MLKL is also connected to clinical prognosis. Interesting conclusions have been drawn from analysis performed in the Affiliated Hospital of Qingdao. The study included samples of normal and cancer colon tissues from 135 patients after radical surgical procedures. The authors divided the samples into two categories with high or low expression of MLKL, based on immunostaining. In the population with a low expression of mixed lineage kinase domain-like protein, the overall survival was shorter (78.6 months vs. 81.2 months; *p* =0.011), even in the subpopulation that received an adjuvant chemotherapy (66.3 vs. 72.9 months; *p* = 0.005), which is also connected with decreased recurrence-free survival (60.4 vs. 72.8 months; *p* = 0.032) [81]. The phosphorylation of MLKL could be induced by chemotherapy with 5-fluorouracil (5-FU) and in that way could lead to a switch to death signaling. The analysis of RNA gene expression, obtained from patients with colorectal cancer, showed that 5-FU does not significantly elevate the cell death process in mucinous cells opposite to non-mucinous cells [82].

A novel therapy based on the intratumor delivery of mRNA is one of the most promising ways to improve prognosis in metastatic colorectal carcinoma. One in vitro study proved that MLKL-mRNA treatment, when combined with immune checkpoint blockades, significantly enhances antitumor activity. MLKL-mRNA treatment rapidly induces T cell responses directed against tumor neoantigens, and requires both CD4+ and CD8+ T cells to prevent tumor growth [83]. We summarized the findings of role of MLKL in CRC in Table 3.

## 7. Other Agents

Necrostatin-1 is a specific inhibitor of necroptosis, preventing the interaction of receptor-interacting protein (RIP) 1 and RIP3. In a study based on mice models of colitis and colitis-associated cancer (CAC), necrostatin-1 significantly suppressed colitis symptoms in mice, including weight loss, colon shortening, colonic mucosa damage and its severity, and the excessive production of interleukin-6. Necrostatin-1 administration inhibited the upregulation of RIP1 and RIP3, and enhanced the expression of caspase-8 in colitis cells. The administration of necrostatin-1 in mouse cells significantly suppressed tumors’ genesis and the development of neoplasm through inhibiting JNK/c-Jun signaling [84].

The glycolipid transfer protein (GLTP) encoded on chromosome 12 (locus 12q24.11) is amphitropic protein which mediates the non-vesicular trafficking of glycosphingolipids and their homeostatic levels within cells. As a result of this, the cell could change its shape, which could be a phenotypic indicator of programmed cell death processes. The overexpression of GLTP inhibits the growth of human colon carcinoma cells (HT-29; HCT-116) but spares normal colonic cells (CCD-18Co). GLTP overexpression arrests the cell cycle at the G1/S checkpoint via the upregulation of cyclin-dependent kinase inhibitor-1B (Kip1/p27) and cyclin-dependent kinase inhibitor 1A (Cip1/p21) at the protein and mRNA levels, and via the downregulation of cyclin-dependent kinase-2 (CDK2), cyclin-dependent kinase-4 (CDK4), and cyclin E and cyclin D1 protein levels. The HT-29 cells underwent cell death by necroptosis, as revealed by phosphorylation of human mixed lineage kinase domain-like protein (pMLKL) via RIPK-3, elevated cytosolic calcium, and plasma membrane permeabilization by pMLKL oligomerization. The overexpression of W96A-GLTP, an ablated GSL binding site mutant, failed to arrest the cell cycle or induce necroptosis. The depletion of RIPK-3 or MLKL abrogated necroptosis induced by GLTP [85].

Protein arginine N-methyltransferase 1 (PRMT1) can methylate RIPK3 at the amino acid of R486 in humans and the conserved amino acid R479 in mice. The methylation of RIP3 by PRMT1 inhibited the interaction of RIP3 with RIP1. On the contrary, the methylation-deficiency RIP3 mutant promoted necroptosis as a basic death pathway. Consequently, the increased infiltration by myeloid-derived immune suppressor cells (MDSC), lead to immune escape and colon cancer progression. Moreover, in one study, authors generated a RIP3 R486 di-methylation specific antibody (RIP3ADMA). The protein levels of PRMT1 and RIP3ADMA in cancer samples were positively correlated, and both of them were associated with longer patient survival. This fact reveals that PRMT1 and RIP3ADMA could be the valuable prognosis markers of colon cancer [86].

A huge analysis of the connection between necroptosis-related (NRG) genes and prognosis of CRC has been performed by Tan L. and colleagues. They used Gene Expression Omnibus (GEO) databases and the Cancer Genome Atlas (TCGA) to formulate the novel gene signature (CTSB, PAEP, ARL4C, TAP2, WFS1, BATF2, DUSP27, CXCL9, EPHB2, IRF8, CXCL13, GZMB, APOL6, NLRC5, CXCL10, IRF1, HES6, and PTGDR). The expression of FADD, MLKL, TLR2, PGAM, HMGB1, CXCL 1, TRAF2, and EZH2 was elevated in CRC cells, in contrast to FAS, RIPK1, RIPK3, TLR3, TNFRSF, ALDH2, and NDRG2, which presented increased expression in healthy intestinal cells. The mutations were detected only in 44 samples from CRC patients, with a missense mutation in FAS, RIPK1, TLR3, NR2C2, and TRAF2 as the predominant aberration. The statistical analysis showed that gene signatures could be connected with prognosis, independent from other clinical factors. Furthermore, the association of gene expression with stage, tumor location, and gender has been observed. The high expression of NRG was connected to a more advanced clinical stage, worse clinicopathological grade, higher frequency of deficient DNA mismatch repair (DMR), higher frequency of Kirsten rat sarcoma 2 viral oncogene homolog (KRAS) gene mutation, and lower BRAF mutation. The authors also suggest some connection between NRG expression and response to immunotherapy. Despite the fact of the exhaustive character of analysis, the presented theses must be proven in further analyses [87].

## 8. Potential Necroptotic Agents in Therapy of Colorectal Cancer

Hunting for the new form of therapy leads researchers to explore new intracellular paths, particularly necroptosis. The first potential aim is cationic peroxidase (POD), purified from proso millet seeds (PmPOD). Potential toxicity for normal cells was also observed in neoplasmatic cells lines, especially in HT29 cells (line obtained from Caucasian female with colorectal adenocarcinoma). The exposition of HCT116 (adult male with colon cancer) and HT29 to PmPOD induced interactions between RIPK1 and RIPK3, which consequently initiated necroptosis [88]. Ergothioneine (Egt) is a dietary amino acid, which has also been shown to promote necroptosis through activation of the RIP1/RIP3/MLKL pathway. Immunoprecipitation assays revealed increased interaction between the terminal effector in necroptotic signaling, MLKL, and SIRT3 during Egt therapy. Silencing of the SIRT3 gene blocked the upregulation of MLKL, which could decrease promotion of necroptosis [89].

Ex vivo treatment of HCT116 and HT-29 cell lines by 2-methoxy-6-acetyl-7-methyljuglone (MAM), led to the formation of a complex with RIPK1/RIPK3. As a result, mitochondrial reactive oxygen species’ (ROS) level increased and promoted necroptosis. Furthermore, MAM-induced cell death was attenuated by the pharmacological inhibitors (caspase inhibitor z-vad-fmk or necrosis inhibitor 2-1H-Indol-3-yl-3-pentylamino-maleimide -IM54) or genetical blockage (necrostatin-1 and siRNA-mediated gene silencing) [90]. Cobalt chloride also demonstrated potential for the initiation of necroptosis in cell line HT-29 [91]. Another course involves the sensitization of resistant colorectal cancer cells to chemotherapy—for example, 5-FU by the pan-caspase inhibitor IDN-7314. This enzyme leads to 5-FU-induced necroptosis, mediated by autocrine secretion of tumor necrosis factor ƒΏ (TNF-ƒΏ). Effectiveness was proven by the decrease in tumor growth in vitro [92]. One of the major challenges in oncological practice is the multidrug resistance of neoplasmatic cells, which is often connected with non-apoptotic programmed cell death (PCD) pathways such as necroptosis. Glycogen synthase kinase 3 (GSK3) takes part in regulating drug-resistance- and chemotherapy-induced necroptosis. In vitro models suggest that inhibition of only one isoform (A or B), or rather the partial inhibition of overall cellular GSK3 activity, is enough to re-sensitize drug-resistant cells to chemotherapy [93]. In another study, Chloroquine (CQ) was found to significantly upregulate receptor-RIPK3 expression. Overexpressed eGFP-RIP3 co-localized with the selective autophagy receptor p62. CQ induced lysosomal membrane permeabilization (LMP) and necroptosis in cancer cells, leading to cancer cell death. Chloroquine is a potential crushing agent for drug resistance [94]. We summarize potential therapeutic agents in the therapy of colorectal cancer in Table 4.

## 9. Conclusions

Necroptosis represents a promising path for researchers to find the trigger point in colorectal cancer therapy. Multiple crucial points and connections allow the extension of the opportunities to understand deeper the process of necroptosis in colorectal cancer cells, and indicate novel potential therapeutic paths. Before we completely understand the role of necroptosis in colorectal cancer, some key challenges have to be explained. Firstly, necroptosis leads to a massive release of DAMPs, which are associated with inflammatory response. Reports suggest the role of DAMPs in prognosis of CRC. Authors have tried to build some predictive models; however, further research is still required to confirm that fact [95]. Furthermore, tumor heterogeneity could influence the necroptosis process because of the variable expression of necroptotic trigger points in colorectal cells, even in the same tumor. Cellular heterogeneity has led experts to attempt to profile colorectal cancer, however the progress of that branch is insufficient at the moment [96]. The identification of molecular targets is also limited by crosstalk with other programmed death pathways, not only with apoptosis but also autophagy [97]. Potential resistance to necroptosis mechanisms also has to be taken into consideration as a limitation. The expression of RIPK1 and RIPK3 is inhibited by hypoxia, which could turn into resistance to targeted therapy [98].

We should emphasize that the investigations were mostly carried out in vitro, which limited the real properties of necroptosis trigger points as potential therapeutic targets. Admittedly, some of them are extensive, like the dissertation of Kang and colleagues [87], but still have to be confirmed in further projects. The in vivo models are based on mice and are related to the recent discovery of necroptosis’ role in tumorigenesis or metastasis [78], which could be regarded as a limitation of the above attempts. The investigations focused on potential novel therapeutic agents are in their initial phase and cannot be considered as clinical trials, but only directions for the future. Despite that fact, necroptosis could take place in targeted therapy after more profound analysis. Potential future analyses should be based on in vivo studies, patient-derived models, systematic evaluation of combination regimens, and early-phase clinical trials assessing both the efficacy and safety of necroptosis modulators. More advanced studies probably show the relevance and potential of necroptosis trigger points in medical practice.

## 10. Materials and Methods

The literature search was performed using the following electronic databases: PubMed, Scopus, and Web of Science. We included peer-reviewed articles published between 1988 and 2025, written in English, which focused on necroptosis, especially in colon cancer. Both original articles and high-quality narrative or systematic reviews were considered.

Studies were included if they met the following criteria: reviews and original articles concentrating on molecular aspects of necroptosis, clinical significance in colon cancer, and proteins which have been part of the programmed death cell pathway. There were not any exclusion criteria; however, the reports and review had to be of high quality.

## Figures and Tables

**Figure 1 ijms-26-11101-f001:**
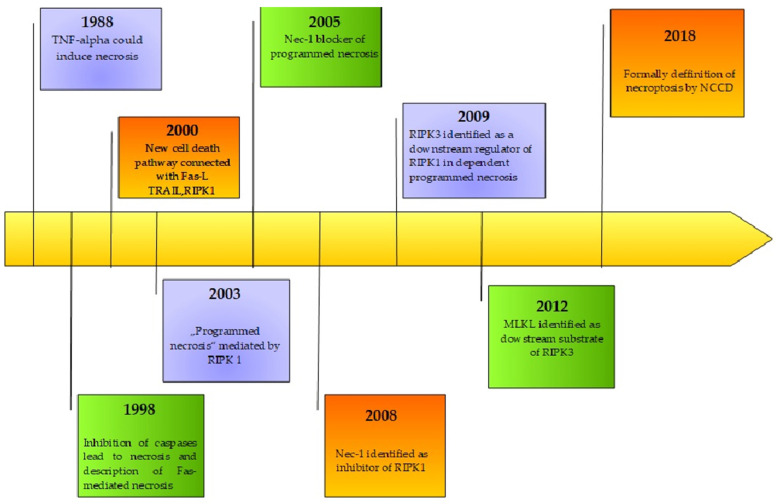
The natural history of necroptosis.

**Figure 2 ijms-26-11101-f002:**
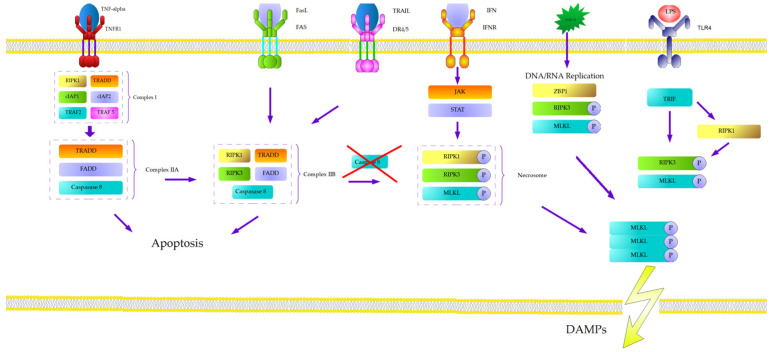
The molecular pathway of necroptosis.

**Table 1 ijms-26-11101-t001:** The findings focused on role of RIPK1 in CRC.

Mechanism	Experimental Model	Therapeutic Implications	Reference
**Interaction with mitochondrial Ca^2+^ uniporter**	**Human CRC cells obtained from patients**	**Promotion of proliferation by increasing the mitochondrial Ca^2+^**	[70]
**Analysis of mRNA transcription of fragile X messenger ribonucleoprotein (FMRP)**	**Human CRC cells obtained from patients**	** The FMRP takes part in controlling RIPK1 expression and necroptotic activation in CRC. **	[71]
**Interaction with TRAF-6**	** SW480 and HCT116 human colon cancer cell lines, MC38 mouse colon cancer, HEK293T cell line **	** TRAF-6 promotes colorectal cell progression by inhibiting the RIPK1/RIPK3/MLKL necroptosis signaling pathway, **	[72]
** RIPK1 in controlling WNT/β-catenin canonical signaling **	** Colorectal cancer cell lines HCT116 and DLD-1 and mice in vivo models **	** RIPK1 plays role in the WNT3A–RIP1–β-catenin pathway in CRC cells (enhancing migration and invasion) **	[73]

**Table 2 ijms-26-11101-t002:** The findings focused on role of RIPK3 in CRC.

Mechanism	Experimental Model	Therapeutic Implications	Reference
** Connection between RIPK3 expression to response for 5-fluorouracil therapy in metastatic CRC **	**Human CRC cells obtained from patients**	** The high expression of RIPK3 is associated with longer (*p* = 0.02) OS (*p* = 0.036) and lower risk of disease progression (*p* = 0.044) **	[75]
** Overcome cell death resistance in caspase-8-deficient colorectal cancer (CRC) ** **and expression of RIPK3**	**Mouse and human cell model**	** Xenograft mouse cell model of caspase-8-deficiency leads to regression of tumors. ** RIPK3 is highly expressed in mouse models of CRC and in a subset of human CRC cell lines	[76]
**Role of** ** RIPK3-deficiency in tumorogenensis by uncontrolled activation of NF-κB, STAT3, AKT, and Wnt-β-catenin ** **pathways**	** Human CRC cells obtained from patients **	** The expression of RIPK3 is reduced in tumors from patients with inflammatory bowel diseases. ** The expression of RIPK3 is downregulated in human CRC and correlated with cancer progression.	[77]
** Role of RIPK3 in the progression of colitis-associated cancer (CAC) **	** Human CAC cells obtained from patients and mouse CRC cells **	** RIPK 3 expression was upregulated in mouse CAC and human colon cancer ** High expression of RIPK3 enhances the proliferation of premalignant intestinal epithelial cells (IECs) and promotes myeloid-cell-induced adaptive immune suppression	[78]

**Table 3 ijms-26-11101-t003:** The findings focused on role of MLKL in CRC.

Mechanism	Experimental Model	Therapeutic Implications	Reference
** Role of MLKL in colitis and colitis-associated tumorigenesis **	**In vivo mouse model**	** MLKL plays a role in maintaining intestinal homeostasis, protecting against colitis and tumorigenesis. **	[81]
**Association between expression of MLKL and clinical prognosis**	** Human CRC cells obtained from patients **	** The low expression of MLKL is connected with shorter OS (*p* = 0.011), even in the subpopulation that received an adjuvant chemotherapy *p* = 0.005), and shorter PFS (*p* = 0.032) **	[82]
** Attempt to determine expression of apoptosis and necroptosis mediators in mucinous CRC by evaluating RNA gene expression **	** Mouse ** ** CRC cell lines-SW1463 (mucinous rectal), SW837 (non-mucinous rectal), LS174T (mucinous colon) and HCT116 (non-mucinous colon) **	** Treatment with 5-FU did not significantly elevate cell death events in mucinous cells, while non-mucinous cells showed robust cell death responses. ** 5-fluorouracil-induced phosphorylation of MLKL in mucinous cancer cells.	[83]
** Generic antitumor therapy based on the intratumor delivery of mRNA ** ** encoded MLKL **	** Mouse cell lines **	** Inhibition of primary tumor growth and protects against distal metastasis. ** Improve antitumor activity in combination with i mmune checkpoint inhibitor	[84]

**Table 4 ijms-26-11101-t004:** Potential therapeutic target/agents in the therapy of colorectal cancer connected with necroptosis.

Potential Therapeutic Target/Agent	Trigger Point	Reference
**PmPOD (Proso millet Cationic Peroxidase)**	**RIPK1, RIPK3**	[88]
**EGT (Ergothioneine)**	**RIPK1/RIPK3/MLKL and SIRT 3**	[89]
**MAM (2-methoxy-6-acetyl-7-methyljuglone)**	**RIPK1/RIPK3**	[90]
**Cobalt chloride**	**Unknown**	[91]
**Pan-caspase inhibitor IDN-731**	**ƒΏ (TNF-ƒΏ)**	[92]
**GSK3 -glycogen synthase kinase 3**	**Unknown**	[93]
**CQ-** **Chloroquine**	**RIPK3**	[94]

## Data Availability

No new data were created or analyzed in this study. Data sharing is not applicable to this article.

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
