# Peer review of "Trigger Points of Necroptosis (RIPK1, RIPK3, and MLKL)—Promising Horizon or Blind Alley in Therapy of Colorectal Cancer?"

_ijms, 2025, doi:10.3390/ijms262211101_

Round 1

Reviewer 1 Report

Comments and Suggestions for Authors

This review article by Sokołowski and Butrym provides a summary of current research exploring novel therapeutic targets to treat colorectal cancers. The review article describes the molecular mechanism involved in necroptosis and highlights the role of three major proteins involved in initiating necroptosis and their potential role as therapeutic targets. Additionally, the review also provides a summary of other protein targets in consideration for modulation of necroptosis as well as current drugs being tested.

There are several major concerns that need to be addressed.

  • In some instances, the language used is very similar to the original text being referenced.

Instance 1: “such as reduction in smoking and increased use of nonsteroi- dal anti-inflammatory drugs, and the uptake of CRC screening among individuals aged 50 years and older”

Instance 2: “In population with colorectal cancer, low expression of mixed lineage kinase domain-like protein has been associated with decreased overall survival (78.6 months vs 81.2months; P = .011). Furthermore, in subpopulation who received adjuvant chemother- apy, low expression of mixed lineage kinase domain-like protein was also connected with decreased recurrence-free survival (60.4 vs 72.8 months; P = .032) and overall survival (66.3 vs 72.9 months; P = .005)”

Additionally, no context for this study is provided. In the original manuscript cited here, the authors describe categorizing the collected samples into high and low expression of MLKL based on immunostaining. A more thorough description of the cited studies and the relevant context need to be included.

Instance 3: “However, chemotherapy with 5-fluorouracil (5-FU) does not significantly elevate cell death process in mucinous cells, whereas non-mucinous cells showed robust cell death responses.”

Instance 4: “the methylation-deficiency RIP3 mutant promoted necroptosis, immune escape and colon cancer progression due to increasing tumor infiltrated myeloid-derived immune suppressor cells (MDSC), while PRMT1 reverted the immune escape of RIP3 necroptotic colon cancer”

The authors need to paraphrase the referenced text and not use verbatim language.

  • There is a general lack of readability with several errors in language and punctuation. In several instances, the appropriate context is not provided.

“RIPK1 is a crucial enzyme involved in the necroptosis and apoptosis process [64]. In the human genome, it’s encoded by the RIPK1 gene, which is located on chromosome”: Incomplete sentence, does not mention which chromosome.

“Overexpression of RIPK 1 was also observed in mouse cell models of colorectal cancer (CRC) and in a subset of human CRC cell lines [76].”

This statement appears under the section describing protein RIPK3 but is incorrectly identified as RIPK1. The cited text also mentions RIPK3.

“In another in vitro study, upregulation of RIPK3 in mouse colitis-associated cancer cells and in human colon cancer cells deficient in RIPK3 in mice line significantly attenuated colitis-associated tumorigenesis. The authors suggested two RIP3-induced progression path - JNK (c-Jun N-terminal kinase ) and CXCL1 ( chemokine (C-X-C motif) ligand 1 ) signaling”.

Relevant context and observations from the original study is not correctly described.

Language errors:

Instance 1: “Consecutive researches focus on novel agents and new pathways of therapy. They are crucial to extend overall survival patients with advanced CRC”: Language errors

Instance 2: “Beside rapid progress of cancer-directed surgery and new therapy agents for ad- vanced colorectal cancer, treatment effects are insufficient, what leads to looking forward to new pathway and trigger points.”

Instance 3: “The next step was the observation of downstreaming substrate of RIPK-3 in TNF -induced “programmed necrosis “ , the mixed lineage kinase domain-like pseudokinase (MLKL)

“anty-EGFR”: Typographical errors 

  • There seem to be several incorrect references.

References 12 and 33 are the same and have been repeated twice in the bibliography and cited twice in text.

“Cooperation between RIPK1 and RIPK3 leads to activation of the latter and formation of the protein Complex II (TRADD, FADD- Fas - associated protein with death domain, and caspase-8) [38]” : The references cited here do not correspond to the text.

“regarofenib [9], new chemotherapy agent - trifluridine and tipiracil hydrochloride [10] and immune checkpoint inhibitors like pembrolizumab [11] or nivolumab with ipilimumab [12]” : The references cited here do not correspond to the text.

“To shift a balance to necroptosis, two independent factors have to be fulfilled, The first one is a blockade of caspase 8 – generally genetically [40] or pharmacologically by a pan-caspase inhibitor Z-VAD-fmk [41]”: The references cited here do not correspond to the text.

  • It would be useful if the authors include a summary table or a schematic that shows all the proteins involved in necroptosis currently in consideration as therapeutic targets.

  • There are spelling errors in the Figure 1 schematic (“downstream”, “definition”) and in the title for Figure 2 (“The molecural pathaway onf necroptosis is presented on Figure 2.”)

  • There are some recent articles that further describe the role of RIPK1 in colorectal cancers and a recent study that examines expression of necroptosis genes in colorectal cancers. The authors should update the manuscript with these latest studies.

(PMID: 36604411, PMID: 38681991, PMID: 37500894, PMID: 40959081)

Comments on the Quality of English Language

There is a general lack of readability and several language and typographical errors in the manuscript. The authors should address the language and punctuation issues and improve the overall presentation and flow of information. 

Author Response

Dear Reviever,

We want to thank you for detailed reviev of our manuscript. According to your suggestions , we prepared a new version of our article. To In order to facilitate , the next reviev we marked all chages in text on yellow.

Comment 1 : In some instances, the language used is very similar to the original text being referenced.

Responce 1 : We paraprhased some simmilar sentences for example:

Instance 1: “such as reduction in smoking and increased use of nonsteroi- dal anti-inflammatory drugs, and the uptake of CRC screening among individuals aged 50 years and older”

We replace it to : 

"This decline is associated with changing in lifestyle for example restricion in smoking andcommon use of nonsteroidal anti-inflammatory drugs [1] . Futhermore mass colonoscopy screening lead to increased proportion of diagnosis of CRC in localized stage disease. The percentage increased from 33% in 1995 to 41% in 2005 [1]"

Instance 2: “In population with colorectal cancer, low expression of mixed lineage kinase domain-like protein has been associated with decreased overall survival (78.6 months vs 81.2months; P = .011). Furthermore, in subpopulation who received adjuvant chemother- apy, low expression of mixed lineage kinase domain-like protein was also connected with decreased recurrence-free survival (60.4 vs 72.8 months; P = .032) and overall survival (66.3 vs 72.9 months; P = .005)”

Additionally, no context for this study is provided. In the original manuscript cited here, the authors describe categorizing the collected samples into high and low expression of MLKL based on immunostaining. A more thorough description of the cited studies and the relevant context need to be included.

We added context and changed it to:

"Similar to other crucial necroptotic proteins, the expression of MLKL is also connected to clinical prognosis. The interesting concusions have been drawn from analysis performed in The Affiliated Hospital of Qingdao. The study included samples of normal and cancer colon tissues from 135 patients after radical surgical procedures. The authors divided the samples into two categories with high or low expression of MLKL based on immunostaining In general population with low expression of mixed lineage kinase domain-like protein the overall survival was shorter (78.6 months vs 81.2months; P = .011), even in subpulation who receiving an adjuvant chemotherapy (66.3 vs 72.9 months; P = .005), which is also connected wo decreased recurrence-free survival (60.4 vs 72.8 months; P = .032)"

Instance 3: “However, chemotherapy with 5-fluorouracil (5-FU) does not significantly elevate cell death process in mucinous cells, whereas non-mucinous cells showed robust cell death responses.”

We paraprhased it to:

"The phophorylation of MLKL could be induced by chemotherapy with 5-fluorouracil (5-FU) and in that way lead to switch a death signalling. The analysis of RNA gene expression obtained from patients with colorectal cancer, showed that 5-FU does not significantly elevate the cell death process in mucinous cells opposite to non-mucinous cells"

Instance 4: “the methylation-deficiency RIP3 mutant promoted necroptosis, immune escape and colon cancer progression due to increasing tumor infiltrated myeloid-derived immune suppressor cells (MDSC), while PRMT1 reverted the immune escape of RIP3 necroptotic colon cancer”

We have change it to :

„Protein arginine N-methyltransferase 1 (PRMT1) can methylate RIPK3 at the amino acid of R486 in human and the conserved amino acid R479 in mouse. The methylation of RIP3 by PRMT1 inhibited the interaction of RIP3 with RIP1. On the contrary, the methylation-deficiency RIP3 mutant promoted necroptosis as basic death pathway. Consequently, the inceased infiltration by myeloid-derived immune suppressor cells (MDSC), lead to immune escape and colon cancer progression"

Comment 2 :There is a general lack of readability with several errors in language and punctuation. In several instances, the appropriate context is not provided.

Responce 2: We have checked and correct all errors for example :

“RIPK1 is a crucial enzyme involved in the necroptosis and apoptosis process [64]. In the human genome, it’s encoded by the RIPK1 gene, which is located on chromosome”: Incomplete sentence, does not mention which chromosome.

We added: "chromosome 6 on loci 6p25.2."

“Overexpression of RIPK 1 was also observed in mouse cell models of colorectal cancer (CRC) and in a subset of human CRC cell lines [76].”

This statement appears under the section describing protein RIPK3 but is incorrectly identified as RIPK1. The cited text also mentions RIPK3.

We changed it to RIPK3

“In another in vitro study, upregulation of RIPK3 in mouse colitis-associated cancer cells and in human colon cancer cells deficient in RIPK3 in mice line significantly attenuated colitis-associated tumorigenesis. The authors suggested two RIP3-induced progression path - JNK (c-Jun N-terminal kinase ) and CXCL1 ( chemokine (C-X-C motif) ligand 1 ) signaling”.

We have changed a text to :

Other in vitro study, was focused on role of RIPK3 in collitis associated tumorogenesis. The samples has been obtained from tissues from colorectal cancer patients and mice cells, where the colitis-associated cancer  was induced using azoxymethane injection followed by dextran sodium sulfate treatment in RIP3-deficient or wild-type mice cells. The analysis showed that,RIPK3 is upregulated in mouse colitis-associated cancer cells and in human colon cancer cells. The deficient of RIPK3 in mice cells were connected with inhibition of collitis associated tumorogenesis.The crucial role in tumorogensisi by The immunosupresive microenviorment play an expression of chemokine CXCL1, which is induced by RIPK3. In colcusion autors suggest that RIP3 could enhancing the proliferation of premalignant intestinal epithelial cells (IECs) via JNK (c-Jun N-terminal kinase ) signaling. [78].

Comment 3:Language errors

Instance 1: “Consecutive researches focus on novel agents and new pathways of therapy. They are crucial to extend overall survival patients with advanced CRC”: Language errors

We changed it to:

"The novel agents based on new molecular pathways, could definitely change that unfavorable statistics and extend the overall survival in population of patients with advanced CRC"

Instance 2: “Beside rapid progress of cancer-directed surgery and new therapy agents for ad- vanced colorectal cancer, treatment effects are insufficient, what leads to looking forward to new pathway and trigger points.”

We have changed it to:

"Despite the significant progress in therapy of advanced colorectal cancer, the effects of treatment are still insufficient. The contemporary oncology is still looking for new trogger point and prostective branches of therapy."

Instance 3: “The next step was the observation of downstreaming substrate of RIPK-3 in TNF -induced “programmed necrosis “ , the mixed lineage kinase domain-like pseudokinase (MLKL)

We have changed it to:

"Sun L. and collegues described the mixed lineage kinase domain-like pseudokinase (MLKL) as a downstreaming substrate of RIPK-3 in TNF -induced “programmed necrosis“,afterwards."

“anty-EGFR”: Typographical errors 

We have corrected it to :"monoclonal antibodies targeted EGFR"

  • There seem to be several incorrect references.

References 12 and 33 are the same and have been repeated twice in the bibliography and cited twice in text.

We have changed the reference 12 to correct one ;

"Andre T, Elez E, Van Cutsem E et al; CheckMate 8HW Investigators. Nivolumab plus Ipilimumab in Microsatellite-Instability-High Metastatic Colorectal Cancer. N Engl J Med. 2024;28;391(21):2014-2026. doi: 10.1056/NEJMoa2402141"

“Cooperation between RIPK1 and RIPK3 leads to activation of the latter and formation of the protein Complex II (TRADD, FADD- Fas - associated protein with death domain, and caspase-8) [38]” : The references cited here do not correspond to the text.

We correct the reference to:

"Wegner KW, Saleh D, Degterev A. Complex Pathologic Roles of RIPK1 and RIPK3: Moving Beyond Necroptosis. Trends Pharmacol Sci. 2017 r;38(3):202-225. doi: 10.1016/j.tips.2016.12.005."

“regarofenib [9], new chemotherapy agent - trifluridine and tipiracil hydrochloride [10] and immune checkpoint inhibitors like pembrolizumab [11] or nivolumab with ipilimumab [12]” : The references cited here do not correspond to the text.

Responce : We have corrected the citations.

“To shift a balance to necroptosis, two independent factors have to be fulfilled, The first one is a blockade of caspase 8 – generally genetically [40] or pharmacologically by a pan-caspase inhibitor Z-VAD-fmk [41]”: The references cited here do not correspond to the text.

We have changed the refferences to:

"40.Fritsch M, Günther SD, Schwarzer R etal. Caspase-8 is the molecular switch for apoptosis, necroptosis and pyroptosis. Nature. 2019;575(7784):683-687. doi: 10.1038/s41586-019-1770-6.

41.Koike A, Hanatani M, Fujimori K. Pan-caspase inhibitors induce necroptosis via ROS-mediated activation of mixed lineage kinase domain-like protein and p38 in classically activated macrophages. Exp Cell Res. 2019;15;380(2):171-179. doi: 10.1016/j.yexcr.2019.04.027."

It would be useful if the authors include a summary table or a schematic that shows all the proteins involved in necroptosis currently in consideration as therapeutic targets.

We have added suitable table

There are spelling errors in the Figure 1 schematic (“downstream”, “definition”) and in the title for Figure 2 (“The molecural pathaway onf necroptosis is presented on Figure 2.”)

We correct the figures

There are some recent articles that further describe the role of RIPK1 in colorectal cancers and a recent study that examines expression of necroptosis genes in colorectal cancers. The authors should update the manuscript with these latest studies.

(PMID: 36604411, PMID: 38681991, PMID: 37500894, PMID: 40959081)

We have added all suggested articles to our reviev

We hope that after corrections, our manuscript is suitable to your established Journal.

Yours sincerely,

Marcin Sokołowski and Aleksandra Butrym

Reviewer 2 Report

Comments and Suggestions for Authors

The manuscript provides a comprehensive overview of necroptosis pathways and their potential role in colorectal cancer (CRC). The topic is timely and relevant, given the emerging interest in regulated necrosis as both a mechanism of therapeutic resistance and a novel target for anti-cancer therapy. The authors successfully summarize key molecular players such as RIPK1, RIPK3, and MLKL.

However, the manuscript currently lacks the critical evaluation that would distinguish it as a scholarly review rather than a descriptive summary. Specifically, the authors should place stronger emphasis on assessing the strength and limitations of the cited studies, highlight gaps in current understanding, and provide a clearer perspective on future therapeutic prospects. At present, the manuscript reads more like an annotated summary of the literature than a critical synthesis.

Major Comments

  1. The most important shortcoming of the manuscript is the absence of a balanced critical discussion. The authors should clearly address: Limitations of existing experimental models (e.g., cell lines vs. in vivo systems) and translational challenges in targeting necroptosis pathways for therapy. The conclusion section should synthesize these evaluations into a coherent outlook—does current evidence support necroptosis as a therapeutic horizon or a blind alley?
  2. Sections 4, 5, and 6 currently contain basic information that overlaps with Section 3. These should instead focus specifically on the roles of RIPK1, RIPK3, and MLKL in CRC, respectively. Consider summarizing key findings in a table, outlining each protein, its known mechanisms, associated experimental models, and therapeutic implications.
  3. Section 7 lacks consistency. It begins with a description of necrostatin as a compound and then shifts to expression analyses of other regulatory factors. This section should be restructured for logical coherence and thematic focus.
  4. The manuscript would benefit from additional schematic figures describing canonical and non-canonical necroptosis pathways, interactions between necroptotic and apoptotic signaling, the potential roles of necroptosis in CRC tumor biology and immune response. Figures would enhance reader understanding, especially for non-specialists.

Minor Comments

  • Provide the full form of each abbreviation upon first use. Remove unnecessary or redundant abbreviations to improve readability.
  • Line 53: Replace “pathway” with “pathways.”
  • Lines 64–65: Rewrite for clarity and scientific precision:

“Although researchers had long been unable to identify a novel cell death pathway, continued investigation led to the discovery in 2000 by Holler et al. of a distinct mechanism of programmed cell death induced by tumor necrosis factor-α (TNF-α), tumor necrosis factor–related apoptosis-inducing ligand (TRAIL), or Fas ligand (FasL).”

  • Line 78: The term “Necrostatin” should be lowercase — “necrostatin.”
  • Lines 87–89: Revise as:

“It is also worth noting the discovery reported in 2013 that toll-like receptors (TLRs) can induce necroptosis through a TRIF–RIPK3–MLKL signaling axis, functioning as a pathway independent of death receptor activation.”

  • Lines 95–108: For improved precision and readability, consider:

“Tumor necrosis factor-α (TNF-α) binds to its receptor, tumor necrosis factor receptor 1 (TNFR1), which is typically expressed as a trimer on the cell surface. Upon ligand binding, TNFR1 undergoes conformational changes that enable recruitment of adaptor proteins to its cytosolic death domain. These include TRADD, RIPK1, TRAF2, TRAF5, and cIAP1/2, forming the membrane-bound Complex I. Ubiquitination of RIPK1 by cIAP1/2 stabilizes Complex I and facilitates recruitment of the IκB kinase (IKK) complex (IKKα, IKKβ, and NEMO), leading to NF-κB activation and pro-survival signaling. Deubiquitination of RIPK1 promotes Complex II formation, comprising TRADD, FADD, caspase-8, and RIPK1. Interaction between RIPK1 and RIPK3 within this complex can trigger necroptosis, depending on caspase-8 activity and cellular context.”

  • (lines 108+): Suggested rewrite:

“A key regulatory checkpoint in programmed cell death is the balance between apoptosis and necroptosis, largely governed by caspase-8. Active caspase-8 cleaves and inactivates RIPK1 and RIPK3, suppressing necroptosis and promoting apoptosis. Inhibition or loss of caspase-8 activity—either genetically (e.g., CASP8 mutations) or pharmacologically (e.g., Z-VAD-fmk)—shifts this balance toward necroptosis.”

  • Line 142–145: Ensure consistent font and formatting.
  • Lines 337–345: The remaining parts of the template should be removed before submission.

The manuscript requires careful language editing for grammar, flow, and scientific accuracy. Avoid colloquial expressions and overly descriptive sentences.

Comments on the Quality of English Language

The manuscript requires careful language editing for grammar, flow, and scientific accuracy.

Author Response

Dear Reviever,

We want to thank you for detailed reviev of our manuscript. . We want to inform you, that we extend our manucsipt for several refferences.According to your suggestions , we prepared a new version of our aricle. To In order to facilitate , the next reviev we marked all chages in text on yellow.

Major Comments

  1. The most important shortcoming of the manuscript is the absence of a balanced critical discussion. The authors should clearly address: Limitations of existing experimental models (e.g., cell lines vs. in vivo systems) and translational challenges in targeting necroptosis pathways for therapy. The conclusion section should synthesize these evaluations into a coherent outlook—does current evidence support necroptosis as a therapeutic horizon or a blind alley?

Response 1: We converted the concusion to explain some limitations and translational challenges in targeting necroptosis pathways for therapy.

2.Sections 4, 5, and 6 currently contain basic information that overlaps with Section 3. These should instead focus specifically on the roles of RIPK1, RIPK3, and MLKL in CRC, respectively. Consider summarizing key findings in a table, outlining each protein, its known mechanisms, associated experimental models, and therapeutic implications.

Response 2: As you suggest we have added the tables summarizing key findings focused on role of RIPK1, RIPK3, and MLKL in CRC. We rearanged some chapter hiovewerwe think we cannot described a role of above kinases without molecular informations include in adequate paragraphs.

  1. Section 7 lacks consistency. It begins with a description of necrostatin as a compound and then shifts to expression analyses of other regulatory factors. This section should be restructured for logical coherence and thematic focus.

Response 3: We reconstructed that section , according to your instructions.

  1. The manuscript would benefit from additional schematic figures describing canonical and non-canonical necroptosis pathways, interactions between necroptotic and apoptotic signaling, the potential roles of necroptosis in CRC tumor biology and immune response. Figures would enhance reader understanding, especially for non-specialists.

Response 4. The all paths of necroptosis are include on figure 2. WE correct some mistake suggest by other reviever

Minor Comments

  • Provide the full form of each abbreviation upon first use. Remove unnecessary or redundant abbreviations to improve readability.

  • Line 53: Replace “pathway” with “pathways.”

  • Lines 64–65: Rewrite for clarity and scientific precision:

“Although researchers had long been unable to identify a novel cell death pathway, continued investigation led to the discovery in 2000 by Holler et al. of a distinct mechanism of programmed cell death induced by tumor necrosis factor-α (TNF-α), tumor necrosis factor–related apoptosis-inducing ligand (TRAIL), or Fas ligand (FasL).”

  • Line 78: The term “Necrostatin” should be lowercase — “necrostatin.”

  • Lines 87–89: Revise as:

“It is also worth noting the discovery reported in 2013 that toll-like receptors (TLRs) can induce necroptosis through a TRIF–RIPK3–MLKL signaling axis, functioning as a pathway independent of death receptor activation.”

  • Lines 95–108: For improved precision and readability, consider:

“Tumor necrosis factor-α (TNF-α) binds to its receptor, tumor necrosis factor receptor 1 (TNFR1), which is typically expressed as a trimer on the cell surface. Upon ligand binding, TNFR1 undergoes conformational changes that enable recruitment of adaptor proteins to its cytosolic death domain. These include TRADD, RIPK1, TRAF2, TRAF5, and cIAP1/2, forming the membrane-bound Complex I. Ubiquitination of RIPK1 by cIAP1/2 stabilizes Complex I and facilitates recruitment of the IκB kinase (IKK) complex (IKKα, IKKβ, and NEMO), leading to NF-κB activation and pro-survival signaling. Deubiquitination of RIPK1 promotes Complex II formation, comprising TRADD, FADD, caspase-8, and RIPK1. Interaction between RIPK1 and RIPK3 within this complex can trigger necroptosis, depending on caspase-8 activity and cellular context.”

  • (lines 108+): Suggested rewrite:

“A key regulatory checkpoint in programmed cell death is the balance between apoptosis and necroptosis, largely governed by caspase-8. Active caspase-8 cleaves and inactivates RIPK1 and RIPK3, suppressing necroptosis and promoting apoptosis. Inhibition or loss of caspase-8 activity—either genetically (e.g., CASP8 mutations) or pharmacologically (e.g., Z-VAD-fmk)—shifts this balance toward necroptosis.”

  • Line 142–145: Ensure consistent font and formatting.

  • Lines 337–345: The remaining parts of the template should be removed before submission.

The manuscript requires careful language editing for grammar, flow, and scientific accuracy. Avoid colloquial expressions and overly descriptive sentences.

Minor Comments

  • Provide the full form of each abbreviation upon first use. Remove unnecessary or redundant abbreviations to improve readability.

  • Line 53: Replace “pathway” with “pathways.”

  • Lines 64–65: Rewrite for clarity and scientific precision:

“Although researchers had long been unable to identify a novel cell death pathway, continued investigation led to the discovery in 2000 by Holler et al. of a distinct mechanism of programmed cell death induced by tumor necrosis factor-α (TNF-α), tumor necrosis factor–related apoptosis-inducing ligand (TRAIL), or Fas ligand (FasL).”

  • Line 78: The term “Necrostatin” should be lowercase — “necrostatin.”

  • Lines 87–89: Revise as:

“It is also worth noting the discovery reported in 2013 that toll-like receptors (TLRs) can induce necroptosis through a TRIF–RIPK3–MLKL signaling axis, functioning as a pathway independent of death receptor activation.”

  • Lines 95–108: For improved precision and readability, consider:

“Tumor necrosis factor-α (TNF-α) binds to its receptor, tumor necrosis factor receptor 1 (TNFR1), which is typically expressed as a trimer on the cell surface. Upon ligand binding, TNFR1 undergoes conformational changes that enable recruitment of adaptor proteins to its cytosolic death domain. These include TRADD, RIPK1, TRAF2, TRAF5, and cIAP1/2, forming the membrane-bound Complex I. Ubiquitination of RIPK1 by cIAP1/2 stabilizes Complex I and facilitates recruitment of the IκB kinase (IKK) complex (IKKα, IKKβ, and NEMO), leading to NF-κB activation and pro-survival signaling. Deubiquitination of RIPK1 promotes Complex II formation, comprising TRADD, FADD, caspase-8, and RIPK1. Interaction between RIPK1 and RIPK3 within this complex can trigger necroptosis, depending on caspase-8 activity and cellular context.”

  • (lines 108+): Suggested rewrite:

“A key regulatory checkpoint in programmed cell death is the balance between apoptosis and necroptosis, largely governed by caspase-8. Active caspase-8 cleaves and inactivates RIPK1 and RIPK3, suppressing necroptosis and promoting apoptosis. Inhibition or loss of caspase-8 activity—either genetically (e.g., CASP8 mutations) or pharmacologically (e.g., Z-VAD-fmk)—shifts this balance toward necroptosis.”

  • Line 142–145: Ensure consistent font and formatting.

  • Lines 337–345: The remaining parts of the template should be removed before submission.

The manuscript requires careful language editing for grammar, flow, and scientific accuracy. Avoid colloquial expressions and overly descriptive sentences.

Response 5: We correct the manucript according whole minor comments

We hope that after corrections, our manuscript is suitable to your established Journal.

Yours sincerely,

Marcin Sokołowski and Aleksandra Butrym

Round 2

Reviewer 1 Report

Comments and Suggestions for Authors

The authors have addressed all the comments. 

Author Response

Dear Reviever,

We want to thank you for detailed reviev of our manuscript and accept all changes.

We hope that after corrections, our manuscript is suitable to your established Journal.

Yours sincerely,

Marcin Sokołowski and Aleksandra Butrym

Reviewer 2 Report

Comments and Suggestions for Authors

In my opinion the text still uses metaphorical or informal phrases like “spread wings in therapeutic exploration,” “undiscovered land,” or “blind alley.” These are stylistically interesting but unsuitable for a scientific manuscript as provided e.g. in conclusion section.

Therefore, the work is still superficial and requires English editing to better reflect thoughts and provide more scientifically sound manuscript.

For example, in conclusions section, the authors mention limitations (mostly in vitro work, reliance on mouse models, early-stage studies), but they don’t summarize what has been established, e.g., the key mechanisms, promising pathways, or how necroptosis could be therapeutically exploited.

In my opinion several key challenges must be addressed before necroptosis can be reliably exploited in therapeutic practice. First, the lack of specificity in necroptosis induction poses a risk of off-target effects and collateral tissue injury, as massive release of DAMPs (damage-associated molecular patterns) may trigger excessive inflammation. Second, tumor heterogeneity significantly influences sensitivity to necroptosis-inducing agents, with variable expression of RIPK1, RIPK3, and MLKL even within the same tumor. Third, incomplete understanding of necroptosis regulation and its cross-talk with other cell death or survival pathways complicates the identification of safe and selective molecular targets. Finally, resistance mechanisms, including downregulation or mutation of necroptotic mediators and hypoxia-driven suppression of RIPK1/RIPK3, limit therapeutic efficacy, though development of MLKL-targeting compounds may help overcome these barriers. Future studies should therefore prioritize the establishment of clinically relevant in vivo and patient-derived models, systematic evaluation of combination regimens, and early-phase clinical trials assessing both efficacy and safety of necroptosis modulators. Comprehensive profiling of necroptosis-related biomarkers could also aid in identifying patient subgroups most likely to benefit.

Furthermore, the abstract sets a goal of determining whether necroptosis is a “promising horizon or a blind alley.” The conclusion and the contents of the manuscript never actually answer this clearly. Also, there’s no strong, actionable summary. What should future studies do next? Which specific gaps exist?

I encourage the authors to provide more in-depth (less superficial analysis ) of their work and provide extensive English editing. 

Nevertheless, the authors have tried and mostly provided necessary changes required in the previous revision. 

Therefore, we should give the authors the opportunity to correct their work in the form of a "minor revision" and leave the latter decision to the academic editor to assess the suitability of this work for publication in IJMS.

Comments on the Quality of English Language

The manuscript requires careful language editing for grammar, flow, and scientific accuracy.

Author Response

Dear Reviever,

We want to thank you for detailed reviev of our manuscript. According to your suggestions , we prepared a new version of our article. To In order to facilitate , the next reviev we marked all changes in text on yellow

Comments 1:" In my opinion the text still uses metaphorical or informal phrases like “spread wings in therapeutic exploration,” “undiscovered land,” or “blind alley.” These are stylistically interesting but unsuitable for a scientific manuscript as provided e.g. in conclusion section. "

Responce 1: We deleted all metaphorical or informal phrases from our manuscript

Comments 2: In my opinion several key challenges must be addressed before necroptosis can be reliably exploited in therapeutic practice. First, the lack of specificity in necroptosis induction poses a risk of off-target effects and collateral tissue injury, as massive release of DAMPs (damage-associated molecular patterns) may trigger excessive inflammation. Second, tumor heterogeneity significantly influences sensitivity to necroptosis-inducing agents, with variable expression of RIPK1, RIPK3, and MLKL even within the same tumor. Third, incomplete understanding of necroptosis regulation and its cross-talk with other cell death or survival pathways complicates the identification of safe and selective molecular targets. Finally, resistance mechanisms, including downregulation or mutation of necroptotic mediators and hypoxia-driven suppression of RIPK1/RIPK3, limit therapeutic efficacy, though development of MLKL-targeting compounds may help overcome these barriers. Future studies should therefore prioritize the establishment of clinically relevant in vivo and patient-derived models, systematic evaluation of combination regimens, and early-phase clinical trials assessing both efficacy and safety of necroptosis modulators. Comprehensive profiling of necroptosis-related biomarkers could also aid in identifying patient subgroups most likely to benefit.

Furthermore, the abstract sets a goal of determining whether necroptosis is a “promising horizon or a blind alley.” The conclusion and the contents of the manuscript never actually answer this clearly. Also, there’s no strong, actionable summary. What should future studies do next? Which specific gaps exist?

Responce 2: We have added all suggested topics decribed by reviever

We hope that after corrections, our manuscript is suitable to your established Journal.

Yours sincerey,

Marcin Sokołowski and Aleksandra Butrym